

# Experimental lung injury induces cerebral cytokine mRNA production in pigs

Jens Kamuf[1], Andreas Garcia Bardon[1], Alexander Ziebart[1], Katrin Frauenknecht[2], Konstantin Folkert[1], Johannes Schwab[1], Robert Ruemmler[1], Miriam Renz[1], Denis Cana[2], Serge C. Thal[1] and Erik K. Hartmann[1]

[1] Department of Anesthesiology, Medical Centre of the Johannes Gutenberg-University, Mainz, Germany
[2] Institute of Neuropathology, Medical Centre of the Johannes Gutenberg-University, Mainz, Germany

## ABSTRACT

**Background**. Acute respiratory distress syndrome (ARDS) is an important disease with a high incidence among patients admitted to intensive care units. Over the last decades, the survival of critically ill patients has improved; however, cognitive deficits are among the long-term sequelae. We hypothesize that acute lung injury leads to upregulation of cerebral cytokine synthesis.

**Methods**. After approval of the institutional and animal care committee, 20 male pigs were randomized to one of three groups: (1) Lung injury by oleic acid injection (OAI), (2) ventilation only (CTR) or (3) untreated. We compared neuronal numbers, proportion of neurons with markers for apoptosis, activation state of Iba-1 stained microglia cells and cerebral mRNA levels of different cytokines between the groups 18 hours after onset of lung injury.

**Results**. We found an increase in hippocampal TNFalpha ($p < 0.05$) and IL-6 ($p < 0.05$) messenger RNA (mRNA) in the OAI compared to untreated group as well as higher hippocampal IL-6 mRNA compared to control ($p < 0.05$). IL-8 and IL-1beta mRNA showed no differences between the groups. We found histologic markers for beginning apoptosis in OAI compared to untreated ($p < 0.05$) and more active microglia cells in OAI and CTR compared to untreated ($p < 0.001$ each).

**Conclusion**. Hippocampal cytokine transcription increases within 18 hours after the induction of acute lung injury with histological evidence of neuronal damage. It remains to be elucidated if increased cytokine mRNA synthesis plays a role in the cognitive decline observed in survivors of ARDS.

Corresponding author
Jens Kamuf, kamuf@uni-mainz.de

## INTRODUCTION

The management of patients with acute respiratory distress syndrome (ARDS) has substantially improved over the years (*Rezoagli, Fumagalli & Bellani, 2017*). However, the late effects of critical care as well as health-related quality of life issues are emerging as a significant challenge for ARDS survivors (*Herridge, 2017*). These patients demonstrate an impaired quality of live and a decline in cognitive function even one year after hospital discharge (*Hopkins et al., 2005*; *Mikkelsen et al., 2012*). The cognitive impairment encompasses global cognition and executive function (*Pandharipande et al., 2013*),
approaching cognition scores similar to those seen in patients with moderate traumatic brain injury or Alzheimer's disease (*Herridge et al., 2016*).

The underlying pathophysiological mechanisms are poorly understood, but are likely complex and multifactorial. Global hypoxia (*Kandikattu et al., 2017*), inflammation (*Dantzer et al., 2008*), oxidative stress (*Popa-Wagner et al., 2013*), and autonomic nervous system dysregulation (*Gonzalez-Lopez et al., 2013*) constitute proposed mechanisms. Of note, even mechanical ventilation alone leads to hippocampal apoptosis and could contribute to the cognitive decline (*Gonzalez-Lopez et al., 2013*).

In an earlier study, we examined neuronal injury and inflammation six hours after onset of ARDS in a porcine model (*Kamuf et al., 2018a*). We found an increase in cerebral cytokine expression in mechanically ventilated pigs that was independent of additional lung injury compared to untreated animals. We concluded that six hours of ventilation may not be enough time for a profound cerebral effect, so we aimed to extend the duration of ARDS. We hypothesize that acute lung injury leads to an upregulation of cerebral cytokine production and neuronal degeneration despite lung protective ventilation and that this effect is partly transferred via serum cytokines. The present study therefore compares cerebral mRNA concentrations of different cytokines to neuronal numbers at 18 h after onset of ARDS in a porcine model.

## MATERIALS & METHODS

The institutional and state animal care committee approved this prospective randomized animal study (Landesuntersuchungsamt Rheinland-Pfalz, Koblenz, Germany; approval number G14-1-077), which was conducted in accordance with international guidelines for the care and use of laboratory animals. The animals used in this study were also part of another study that has not yet been published (approved under the same approval number), the untreated animals served as untreated control animals in a previously published study (*Kamuf et al., 2018a*). All animals were housed, fed and euthanized equally. This manuscript adheres to EQUATOR guidelines.

### Anaesthesia and instrumentation

Twenty healthy male pigs (sus scrofa domesticus, weight: 26–33 kg) were randomized to one of three groups: lung injury by central venous injection of oleic acid (OAI, $n = 8$), ventilation only (CTR, $n = 8$), or untreated animals ($n = 4$). After a local breeder delivered the sedated animals (ketamine 4 mg kg$^{-1}$, azaperon 8 mg kg$^{-1}$ intramuscularly), we establishedan intravenous line. Then, anesthesia was induced and maintained by propofol (8-12 mg kg$^{-1}$ h$^{-1}$) and fentanyl (0.1–0.2 mg kg$^{-1}$ h$^{-1}$). 0.5 mg kg$^{-1}$ atracurium was administered to facilitate orotracheal intubation. The Respirator (Engström Carestation®, GE Healthcare, Germany) was started in pressure-controlled mode with the following settings: tidal volume (V$_t$) of 7 ml kg$^{-1}$, PEEP of five cmH$_2$O, fraction of inspired oxygen (FiO$_2$) of 0.4 and a variable respiratory rate to maintain normocapnia. At this point, the animals of the untreated group were sacrificed in deep anesthesia with a central venous injection of 200 mg propofol and 40 mmol potassium. In all other animals, a balanced electrolyte solution (Sterofundin iso, B. Braun, Germany) was continuously infused at a rate

**Table 1  PEEP/FiO$_2$-setting in this experimental study.**

| FiO$_2$ | 0.4 | 0.4 | 0.5 | 0.5 | 0.6 | 0.7 | 0.7 | 0.7 |
|---------|-----|-----|-----|-----|-----|-----|-----|-----|
| PEEP | 5 | 8 | 8 | 10 | 10 | 10 | 12 | 14 |
| FiO$_2$ | 0.8 | 0.9 | 0.9 | 0.9 | 1.0 | 1.0 | 1.0 | 1.0 |
| PEEP | 14 | 14 | 16 | 18 | 18 | 20 | 22 | 24 |

of 5 ml kg$^{-1}$ h$^{-1}$. Vascular catheters were placed ultrasound-guided: an arterial line, a pulse contour cardiac output catheter (PiCCO, Pulsion Medical Systems, Germany), a central venous line and a 7.5-French introducer for a pulmonary arterial catheter were inserted via femoral vascular access. Respiratory and extended hemodynamic parameters were recorded continuously (Datex S/5, GE Healthcare, Germany). Further respiratory parameters and measurements were recorded by the respirator. Normothermia was maintained by body surface warming.

## Study protocol

Following instrumentation, we set the F$_i$O$_2$ to 1.0 and conducted a lung recruitment maneuver (plateau pressure 40 cmH$_2$O for 10 s). Then, baseline, premorbid parameters were assessed. Lung injury was induced as previously published (*Kamuf et al., 2018b*). 0.1 ml kg$^{-1}$ of oleic acid (cis-9-octadecenoic acid) was dissolved in 20 ml of a saline/blood mixture and injected every three minutes via the central venous line in fractions of 2 ml. The procedure was repeated with another 0.1 ml kg$^{-1}$ after 15 min, if the PaO$_2$/FiO$_2$ was higher than 200 mmHg. After ARDS criteria were fulfilled, the animals were treated according to a standard protocol, which was closely adapted to human ICU treatment as previously published (*Kamuf et al., 2018b*). Respirator settings are as follows: $V_t$ 7 ml kg$^{-1}$, FiO$_2$ and PEEP according to Table 1, with an intended peripheral oxygen saturation of 94–98%. Norepinephrine was administered as needed to maintain stable hemodynamic conditions during the experiments (mean arterial pressure > 65 mmHg). 18 h after manifestation of the lung injury, the animals were killed by injecting 200 mg propofol and 40 mmol potassium.

## Post-mortem analysis

For post-mortem analysis we removed the brain and lungs of the animals after they died. The brain was sectioned in the midsagittal plane, one half was fixed in 4% formaldehyde solution for 3 months, a slice of the frontal cortex and hippocampus were removed from the other hemisphere (*Klein et al., 2016*). Both samples were snap frozen (in liquid nitrogen) for molecular biological analysis. Lung was used for wet/dry ratio (*Kamuf et al., 2017*) and histopathology (*Hartmann et al., 2014*).

Initially, the hippocampus was cut in two mm thick slices. Four tissue slices were then sectioned through the entire specimen and embedded in paraffin. Subsequently, additional 4 $\mu$m thick histological sections were prepared. Preparations were stained with hematoxylin/eosin (HE) and examined with light microscopy to assess the number of HE-stained neurons (*Songarj et al., 2015*) (see Figs. S1–Figs. S6 for representative pictures). Furthermore, we quantified the proportion of damaged neurons, whereby eosinophilic

degeneration and nuclear pyknosis were defined as markers for early neuronal damage. To distinguish between different activation states of the microglia, slices were stained with an antibody against Iba-1 (WAKO, Japan). Iba-1 positive cell numbers/mm$^2$ were counted in one visual field (380 mm$^2$). We further differentiated between microglia without branches, defined as activated, and microglia with three or more branches, defined as resting. Microglia with one or two branches were defined as in-between.

Dependent and non-dependent lung regions were sampled, fixed in formalin for paraffin sectioning and stained with HE. Then we applied a lung injury score according to earlier publications (*Ziebart et al., 2014*). The evaluation included seven different parameters: overdistension, epithelial destruction, inflammatory infiltration, alveolar edema, hemorrhage, interstitial edema, and microatelectasis. Per region, each parameter received a severity grade from zero to five points in four non-overlapping fields of view. In a second step, the extent of each parameter was assessed in a global overview of the entire region. This approach results in an overall score of 175 maximum points per region.

To determine cerebral messenger RNA (mRNA) expression of TNFalpha, IL-6, IL-8 and IL-1beta, we used real-time polymerase chain reaction (RT-PCR, Lightcycler 480 PCR System, Roche Applied Science, Germany) (*Hartmann et al., 2015*). mRNA expression data were normalized against peptidylprolyl isomerase A (PPIA) as control gene.

After ARDS-induction, 6, 12 and 18 h later, blood samples were taken via the cental venous catheter and snap frozen for determination of cytokine levels (IL-6 and TNFalpha) using ELISA kits (Porcine IL-6 Quantikine ELISA, Porcine TNFalpha Quantikine ELISA, R&D Systems, Wiesbaden, Germany) according to the instructions of the manufacturer.

## Statistics

Statistical analyses were performed by One-way ANOVA with post-hoc tests for multiple testing (SNK-Test), except for the analyses of the physiologic data. These were analyzed by Two-way Repeated Measures ANOVA with post-hoc correction (SNK-Test). The results were analyzed and graphed using Sigmaplot$^{\circledR}$ 12.5. A $p < 0.05$ was considered significant.

## RESULTS

Induction of lung injury led to a significant decrease in oxygenation ratio and an increase in peak pressure (P$_{peak}$) at all measured time points compared to control animals (CTR). Furthermore, in animals with lung injury induced by central venous injection of oleic acid (OAI), positive end-expiratory pressure (PEEP), inspiratory fraction of O2 (FiO$_2$) and extravascular lungwater index (EVLWI) were significantly higher at selected timepoints compared to CTR. Tidal volume (V$_T$), end-expiratory CO$_2$ and wet-to-dry ratio showed no difference between the groups.

Heart rate and mean pulmonary arterial pressure (MPAP) were significantly increased in OAI compared to CTR at all measured time points after induction of lung injury. Central venous pressure (CVP), pulmonary capillary wedge pressure (PCWP), cardiac index and norepinephrine dose differed at some time points. Mean arterial pressure (MAP) showed no difference between the groups. Cardiopulmonary characteristics of the animals are provided in Tables 2 and 3 and in Tables S1 and S2.

**Table 2  Pulmonal parameters.** Data shown as mean values and standard deviation.

| | | CTR | OAI | *p*-value |
|---|---|---|---|---|
| PEEP | BLH | $4 \pm 0$ | $4 \pm 0$ | n.s. |
| (cm $H_2O$) | 0 h | $4 \pm 0$ | $6 \pm 2$ | 0.043 |
| | 6 h | $4 \pm 0$ | $9 \pm 2$ | <0.001 |
| | 12 h | $4 \pm 0$ | $7 \pm 3$ | <0.001 |
| | 18 h | $4 \pm 0$ | $7 \pm 3$ | 0.002 |
| $P_{peak}$ | BLH | $15 \pm 2$ | $16 \pm 3$ | n.s. |
| (cm $H_2O$) | 0 h | $14 \pm 1$ | $28 \pm 6$ | <0.001 |
| | 6 h | $15 \pm 2$ | $28 \pm 3$ | <0.001 |
| | 12 h | $16 \pm 2$ | $28 \pm 6$ | <0.001 |
| | 18 h | $17 \pm 2$ | $30 \pm 5$ | <0.001 |
| $V_T$ | BLH | $6 \pm 0$ | $6 \pm 1$ | n.s. |
| (ml/kg) | 0 h | $6 \pm 0$ | $7 \pm 1$ | n.s. |
| | 6 h | $6 \pm 0$ | $7 \pm 1$ | n.s. |
| | 12 h | $6 \pm 0$ | $6 \pm 1$ | n.s. |
| | 18 h | $6 \pm 0$ | $6 \pm 1$ | n.s. |
| $etCO_2$ | BLH | $39 \pm 3$ | $38 \pm 4$ | n.s. |
| (mmHg) | 0 h | $36 \pm 2$ | $37 \pm 4$ | n.s. |
| | 6 h | $37 \pm 4$ | $39 \pm 3$ | n.s. |
| | 12 h | $37 \pm 2$ | $39 \pm 3$ | n.s. |
| | 18 h | $36 \pm 2$ | $39 \pm 4$ | n.s. |
| $FiO_2$ | BLH | $40 \pm 0$ | $40 \pm 0$ | n.s. |
| (%) | 0 h | $100 \pm 0$ | $100 \pm 0$ | n.s. |
| | 6 h | $40 \pm 0$ | $55 \pm 10$ | <0.001 |
| | 12 h | $39 \pm 4$ | $48 \pm 10$ | 0.005 |
| | 18 h | $40 \pm 0$ | $45 \pm 10$ | n.s. |
| EVLWI | BLH | $11 \pm 1$ | $10 \pm 2$ | n.s. |
| (ml/kg) | 0 h | $12 \pm 1$ | $19 \pm 4$ | 0.003 |
| | 6 h | $12 \pm 2$ | $21 \pm 7$ | <0.001 |
| | 12 h | $14 \pm 2$ | $17 \pm 6$ | n.s. |
| | 18 h | $14 \pm 3$ | $17 \pm 5$ | n.s. |
| Oxygenation | BLH | $503 \pm 65$ | $496 \pm 58$ | n.s. |
| Ratio | 0 h | $544 \pm 66$ | $101 \pm 28$ | <0.001 |
| (mmHg) | 6 h | $452 \pm 55$ | $188 \pm 70$ | <0.001 |
| | 12 h | $453 \pm 73$ | $221 \pm 48$ | <0.001 |
| | 18 h | $400 \pm 59$ | $216 \pm 50$ | <0.001 |
| Wet-to-dry Ratio | | $5 \pm 0$ | $6 \pm 1$ | 0.08 |

**Notes.**

Abbreviations: $etCO_2$, endtidal $CO_2$; EVLWI, extravascular lung water index; $FiO_2$, inspiratory fraction of $CO_2$; PEEP, positive endexpiratory pressure; $P_{Peak}$, peak pressure; $V_T$, tidal volume.

**Table 3  Cardiovascular parameters.** Data shown as mean values and standard deviation.

| | | CTR | OAI | $p$-value |
|---|---|---|---|---|
| HR | BLH | $77 \pm 10$ | $77 \pm 15$ | n.s. |
| $(\text{min}^{-1})$ | 0 h | $72 \pm 9$ | $124 \pm 32$ | <0.001 |
| | 6 h | $76 \pm 15$ | $119 \pm 40$ | 0.004 |
| | 12 h | $75 \pm 18$ | $117 \pm 39$ | 0.005 |
| | 18 h | $68 \pm 8$ | $112 \pm 44$ | 0.003 |
| MAP | BLH | $72 \pm 5$ | $75 \pm 11$ | n.s. |
| (mmHg) | 0 h | $76 \pm 8$ | $73 \pm 6$ | n.s. |
| | 6 h | $74 \pm 8$ | $67 \pm 8$ | n.s. |
| | 12 h | $71 \pm 12$ | $69 \pm 6$ | n.s. |
| | 18 h | $68 \pm 8$ | $67 \pm 7$ | n.s. |
| MPAP | BLH | $13 \pm 4$ | $16 \pm 3$ | n.s. |
| (mmHg) | 0 h | $12 \pm 3$ | $38 \pm 4$ | <0.001 |
| | 6 h | $14 \pm 4$ | $30 \pm 5$ | <0.001 |
| | 12 h | $16 \pm 2$ | $27 \pm 4$ | <0.001 |
| | 18 h | $15 \pm 2$ | $27 \pm 2$ | <0.001 |
| CVP | BLH | $6 \pm 2$ | $6 \pm 3$ | n.s. |
| (mmHg) | 0h | $6 \pm 3$ | $7 \pm 4$ | n.s. |
| | 6 h | $8 \pm 4$ | $8 \pm 3$ | n.s. |
| | 12 h | $6 \pm 2$ | $9 \pm 3$ | n.s. |
| | 18 h | $6 \pm 1$ | $10 \pm 3$ | n.s. |
| PCWP | BLH | $7 \pm 1$ | $8 \pm 1$ | n.s. |
| (mmHg) | 0 h | $7 \pm 2$ | $9 \pm 2$ | n.s. |
| | 6 h | $7 \pm 1$ | $8 \pm 2$ | n.s. |
| | 12 h | $7 \pm 2$ | $9 \pm 2$ | n.s. |
| | 18 h | $7 \pm 2$ | $10 \pm 1$ | n.s. |
| CI | BLH | $3.41 \pm 0.43$ | $3.34 \pm 0.85$ | n.s. |
| $(\text{l/min/m}^2)$ | 0 h | $3.26 \pm 0.43$ | $3.87 \pm 0.75$ | n.s. |
| | 6 h | $3.40 \pm 0.68$ | $3.87 \pm 0.92$ | n.s. |
| | 12 h | $3.61 \pm 0.61$ | $4.44 \pm 1.41$ | n.s. |
| | 18 h | $3.28 \pm 0.34$ | $4.83 \pm 1.39$ | 0.002 |
| Norepinephrine | BLH | $0.04 \pm 0.11$ | $0.00 \pm 0$ | n.s. |
| ($\mu$g/kg/min) | 0 h | $0.00 \pm 0$ | $0.55 \pm 0.61$ | n.s. |
| | 6 h | $0.01 \pm 0.04$ | $0.58 \pm 1.13$ | n.s. |
| | 12 h | $0.03 \pm 0.05$ | $0.66 \pm 1.02$ | n.s. |
| | 18 h | $0.01 \pm 0.04$ | $1.09 \pm 1.44$ | 0.007 |

**Notes.**

Abbreviations: CI, cardiac index; CVP, central venous pressure; HR, heart rate; MAP, mean arterial pressure; MPAP, mean pulmonary arterial pressure; PCWP, pulmonary capillary wedge pressure.

There was no difference in the number of neurons ($p = 0.69$; Fig. 1) in the hippocampus (pooled data of Gyrus Dentatus, CA1, CA2, CA3 and CA4) between the groups. Histologic evaluation of neuronal damage showed a significantly higher proportion of damaged neurons in the hippocampus of OAI animals compared with untreated animals ($p = 0.02$;

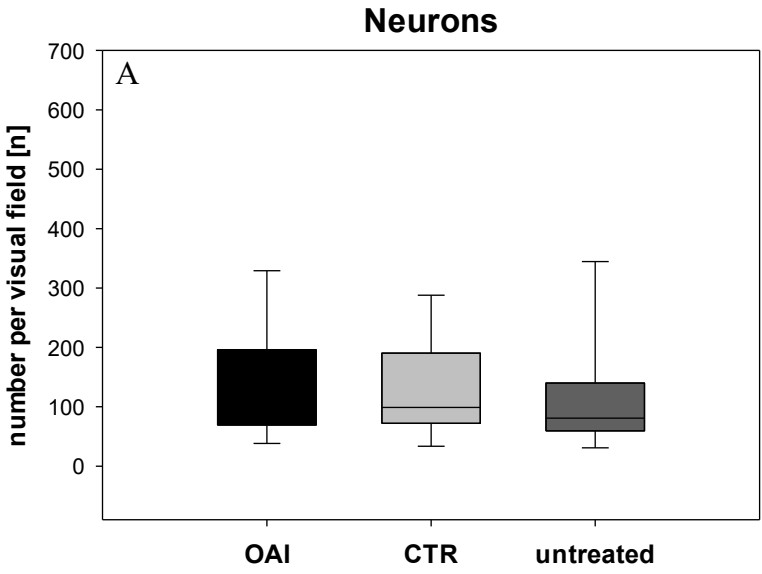

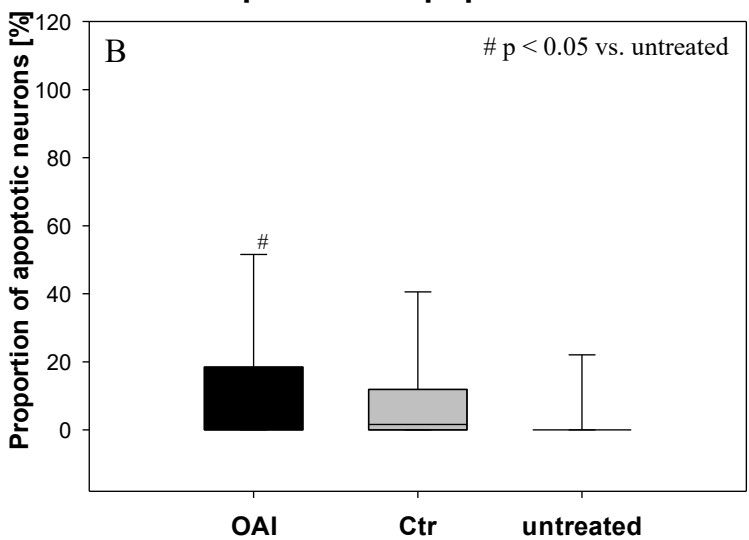

**Figure 1** **Brain histology—neurons.** (A) Mean number of HE-stained neurons in Gyurs dentatus, CA1, CA2, CA3 and CA4 of the Hippocampus. (B) Proportion of apoptotic neurons in Gyurs dentatus, CA1, CA3 and CA4 of the Hippocampus.

pooled data of Gyrus Dentatus, CA1, CA3 and CA4). However, there was no difference between OAI and CTR ($p = 0.11$) or between CTR and untreated ($p = 0.16$; Fig. 1).

The number of Iba-1 stained microglia cells in the hippocampus didn't differ between the groups ($p = 0.71$; Fig. 2), but evaluation of their activation state by counting the number of branches revealed significantly more active microglia cells in OAI and CTR compared to untreated ($p < 0.001$ each), while there was no significant difference between OAI and CTR ($p = 0.12$; Fig. 2). Comparison of resting microglia cells, defined as microglia cells

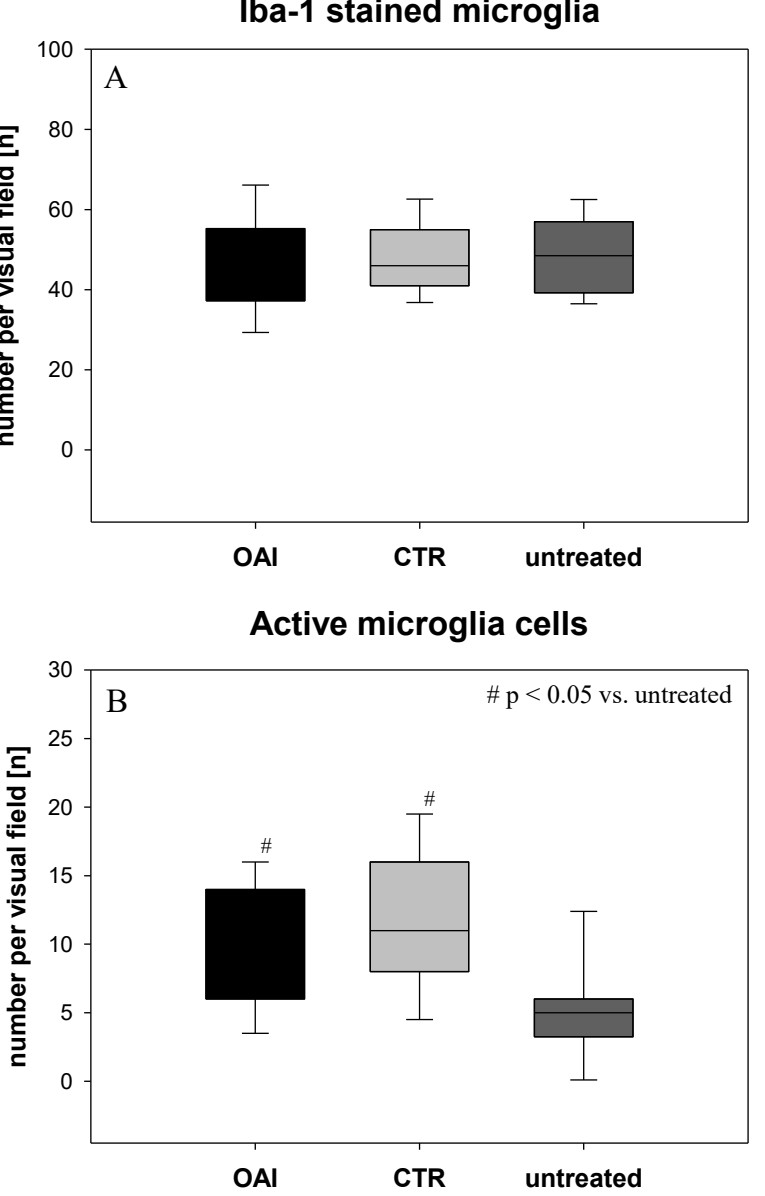

**Figure 2** **Brain histology—microglia cells.** (A) Mean number of Iba-1 stained microglia cells in Gyurs dentatus, CA1, CA2, CA3 and CA4 of the Hippocampus. (B) Mean number of active microglia cells in Gyurs dentatus, CA1, CA3 and CA4 of the Hippocampus.

with three or more branches, showed significantly fewer resting microglia cells in OAI compared to untreated ($p = 0.006$) and in CTR compared to untreated ($p = 0.007$). There was no difference between OAI and CTR ($p = 0.53$).

Lung injury was pronounced in the OAI group ($p = 0.01$ versus CTR, $p < 0.001$ versus untreated). The lungs of the CTR animals showed a significant higher damage score than the untreated group ($p = 0.04$; Fig. 3).

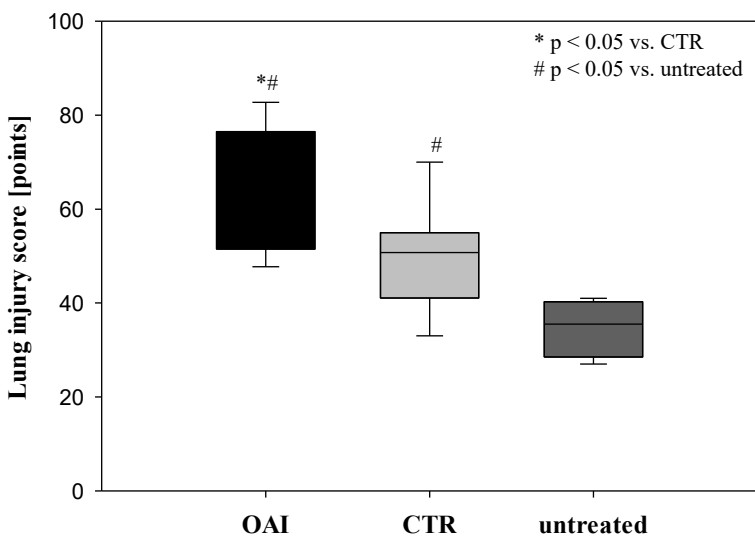

**Figure 3 Lung injury score.** In HE-stained lung slices seven different parameters (overdistension, epithelial destruction, inflammatory infiltration, alveolar edema, hemorrhage, interstitial edema, and microatelectasis) were evaluated with a maximum score of 175 points.

We found no difference in mRNA concentrations of TNFalpha ($p = 0.31$; Fig. 4), IL-6 ($p = 0.062$; Fig. 5), IL-8 ($p = 0.68$; Fig. 6) or IL-1beta ($p = 0.43$; Fig. 7) in the cortex samples. The hippocampus showed a significant increase of TNFalpha mRNA in OAI ($p = 0.005$) and CTR ($p = 0.004$) animals compared to untreated animals. There was no difference between the CTR and OAI group ($p = 0.67$; Fig. 4). IL-6 mRNA was significantly increased in the hippocampus of the animals of the OAI group compared to CTR ($p = 0.02$) and untreated ($p = 0.049$). There was no significant difference between CTR and untreated ($p = 0.62$; Fig. 5). Hippocampal IL-8 ($p = 0.18$; Fig. 6) and IL-1beta ($p = 0.58$; Fig. 7) mRNA copies showed no differences between the groups.

Plasma IL-6 levels in OAI animals were significantly higher after 18 h compared to 12 h ($p = 0.03$) and compared to 6 h ($p = 0.03$; Fig. 8). Plasma IL-6 levels in CTR animals showed no significant difference over time ($p = 0.35$). Plasma TNFalpha levels were significantly higher after ARDS-induction in OAI animals compared to 6 h ($p = 0.02$), 12 h ($p = 0.02$) and 18 h ($p = 0.01$; Fig. 8). There was no significant change in Plasma TNFalpha in CTR animals over time ($p = 0.68$).

## DISCUSSION

In this study, we found an increase in cerebral cytokine mRNA without histological evidence of neuronal damage after OAI-induced lung injury, already within 24 h.

Central venous injection of oleic acid is an established animal model of acute lung injury, which is characterized by alveolar hemorrhage, intravascular thrombosis, infiltration of polymorphonuclear leukocytes, and increased pulmonary microvascular permeability. These changes lead to severe ventilation/perfusion mismatch and increased shunt resulting in hypoxemia and increased mean airway pressures. Furthermore, pulmonary edema

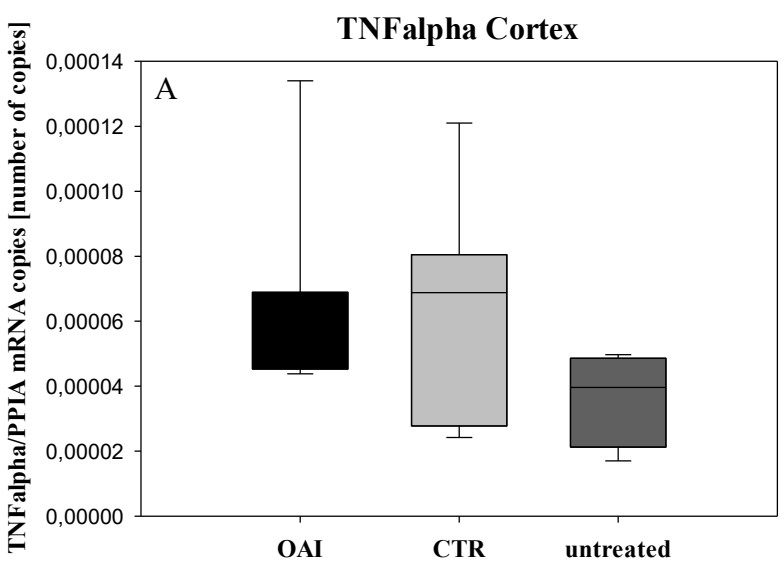

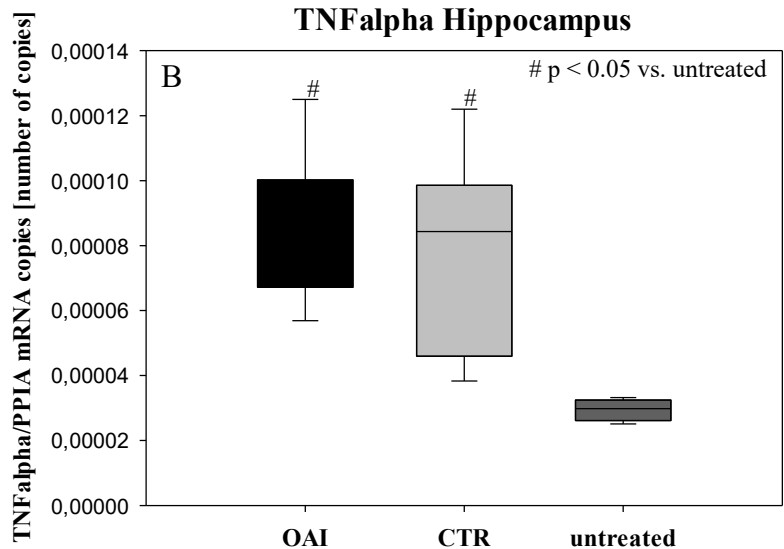

**Figure 4 Cerebral levels of TNFalpha mRNA.** (A) Number of TNFalpha mRNA-copies normalized to PPIA in the cortex. (B) Number of TNFalpha mRNA-copies normalized to PPIA in the hippocampus.

develops, which is characterized by elevation of extravascular lung water and leakage of protein-rich fluid into the airspace and interstitium. Commonly observed hemodynamic effects encompass myocardial depression, early systemic hypotension, and pulmonary hypertension (*Matute-Bello, Frevert & Martin, 2008*), which have been reliably reproduced in our porcine model. In our study, gas exchange and pulmonary arterial hypertension resolved over time in contrast to Derks et al., who reported a maximum effect after 12 h (*Derks & Jacobovitz-Derks, 1977*). This discrepancy may be attributable to continuous lung protective ventilation, which may minimize alveolar stress as well as cyclic recruitment

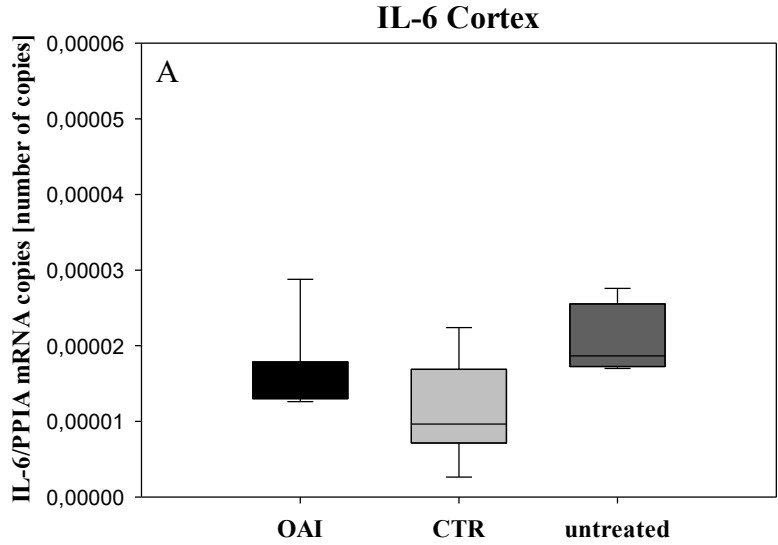

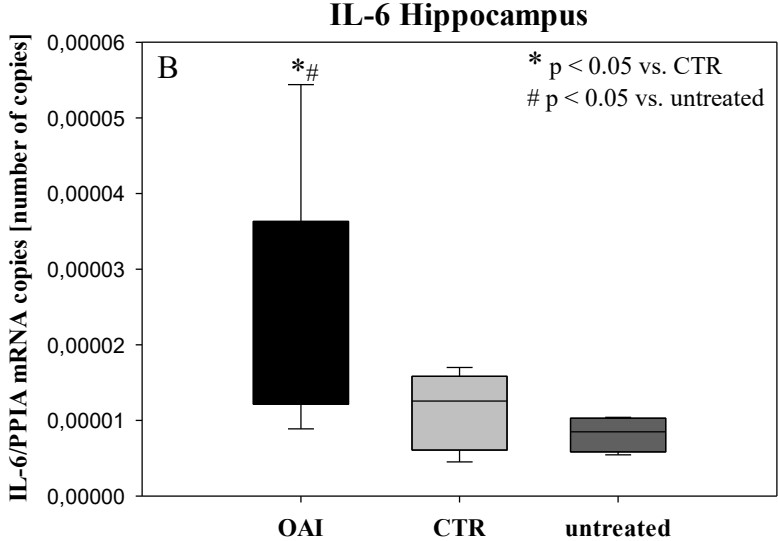

**Figure 5 Cerebral levels of IL-6 mRNA.** (A) Number of IL-6 mRNA-copies normalized to PPIA in the cortex. (B) Number of IL-6 mRNA-copies normalized to PPIA in the hippocampus.

and derecruitment (*Nieman, Gatto & Habashi, 2015*), thereby impeding the inflammatory response.

IL-6 and TNFalpha are important mediators in the development of ARDS (*Goncalves-de Albuquerque et al., 2015*). An increase in serum IL-6 is associated with the development of ARDS (*Aisiku et al., 2016*) and elevated intracranial pressure (*Hergenroeder et al., 2010*) in patients with traumatic brain injury. An increase in these cytokines is also associated with postoperative delirium and cognitive dysfunction (*Wang, Chen & Wu, 2015*). Therefore, we examined systemic levels of IL-6 in the OAI animals immediately after ARDS-induction and found a significant increase, which diminished over time. In OAI animals, only a

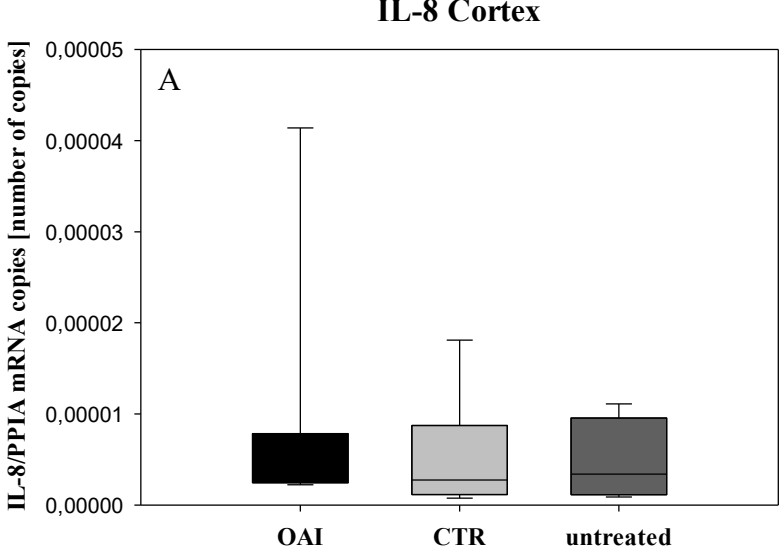

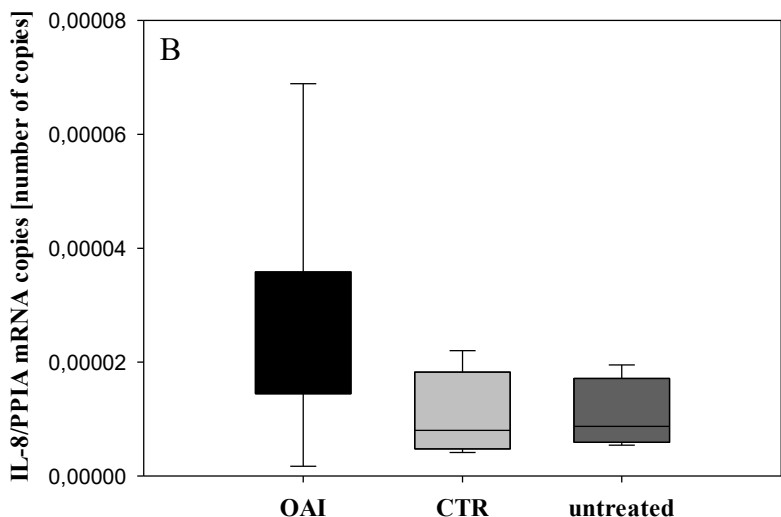

**Figure 6** **Cerebral levels of IL-8 mRNA.** (A) Number of IL-8 mRNA-copies normalized to PPIA in the cortex. (B) Number of IL-8 mRNA-copies normalized to PPIA in the hippocampus.

trend towards higher levels of TNFalpha was observed. This finding is concordant with an animal study on acid aspiration induced lung injury over 5 days, which reported a similar TNFalpha temporal evolution with a tendency to an increase in serum IL-6 (*Bickenbach et al., 2011*).

Circulating TNFalpha can compromise the blood brain barrier (*Tsao et al., 2001*) and activate microglia cells. Importantly, circulating IL-6 is thought to impair the blood brain barrier likewise (*Rochfort & Cummins, 2015*). Furthermore, TNFalpha can activate microglia cells through receptors within the cerebral blood vessels (*Thibeault, Laflamme &*

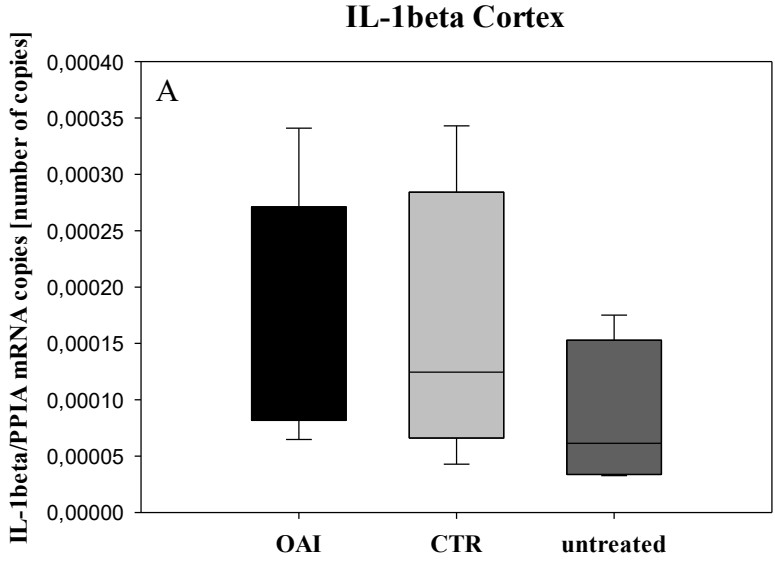

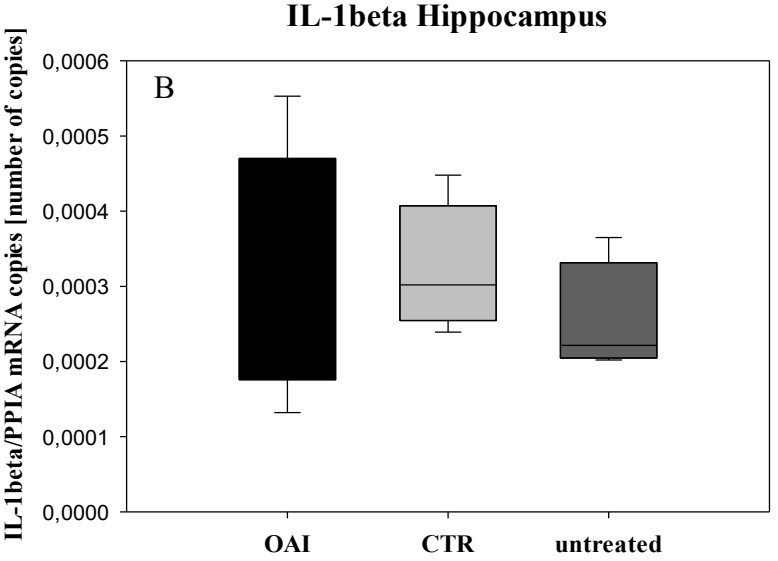

**Figure 7** **Cerebral levels of IL-1beta mRNA.** (A) Number of IL-1beta mRNA-copies normalized to PPIA in the cortex. (B) Number of IL-1beta mRNA-copies normalized to PPIA in the hippocampus.

*Rivest, 2001*). To examine a potential activation of microglia, we assessed the number of iba-1 stained microglia cells, but found no difference between the groups. However, the homogenous distribution of microglia in our present study does not necessarily rule out microglia activation since microglia with similar appearance can exhibit different functional phenotypes, dependent on their microenvironment (*Wes et al., 2016*). Hence, we examined different activation states of microglia cells by counting their branch numbers. Significantly more active microglia cells in OAI and CTR compared to untreated ($p < 0.001$ each) was

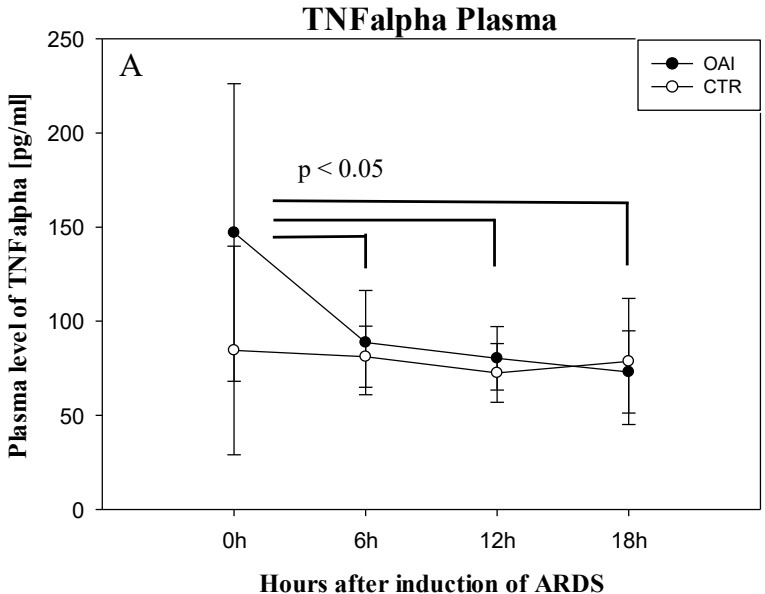

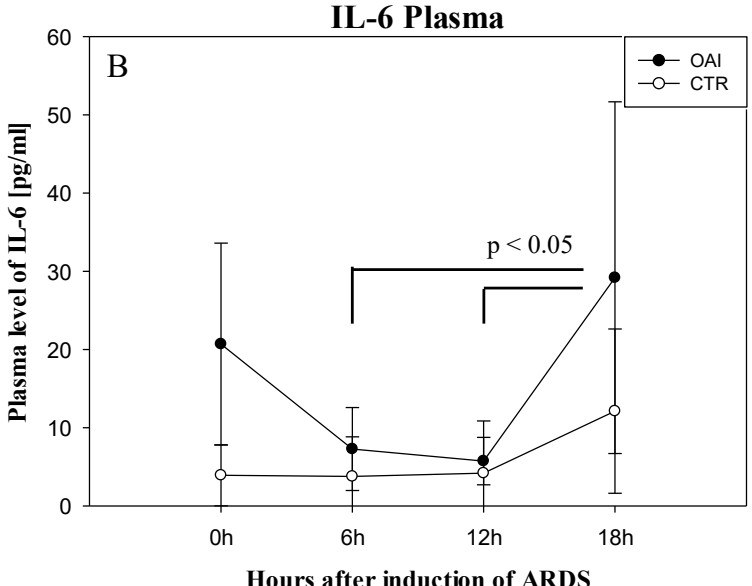

**Figure 8 Time course of plasma cytokines after induction of ARDS.** (A) Plasma levels of TNFalpha over time. (B) Plasma levels of IL-6 over time.

paralleled by significantly fewer resting cells in OAI and CTR compared to untreated (OAI vs. untreated $p = 0.006$; CTR vs. untreated $p = 0.007$). In septic mice an activation of microglia is usually found after 8–48 h (*Hoogland et al., 2015*). Sepsis leads to rapid breakdown of the blood brain barrier causing cerebral inflammation (*Danielski et al., 2018*), but Cytokine release in ARDS is not as pronounced as in sepsis (*Ziebart et al., 2014*). Accordingly, the breakdown—or potential breakdown—of the blood brain barrier may

take longer, which could explain our results. Furthermore, activation of microglial cells is not necessarily accompanied by an increase in numbers. Additional evidence of microglial activation is the cerebral production of TNFalpha or IL-6 mRNA, since these cytokines are mainly released by microglia cells (*Hoogland et al., 2015*) and a hippocampal increase in these cytokines is associated with postoperative cognitive decline (*Chen et al., 2015*). In animals of the OAI group hippocampal IL-6 mRNA was significantly increased compared to untreated and CTR animals. This confirms the results of Bellaver et al., who examined cerebral cytokine production in animals with cecal ligation and perforation. They found an increase in hippocampal IL-6 production (*Bellaver et al., 2018*). Hippocampal TNFalpha mRNA showed a non-significant tendency towards higher level in OAI compared to untreated, but not between OAI and CTR. This finding differs from the results reported by Ruiz-Valdepenas et al., who found a significant increase in cerebral TNFalpha mRNA after lipopoylsaccharide-induced sepsis (*Ruiz-Valdepenas et al., 2011*).

In the next step, cerebral injury was assessed by evaluating neuronal counts in the hippocampus. However, we found no difference between the groups. In contrast, Gonzales-Lopez et al. described initial signs of neuronal apoptosis in mechanically ventilated mice after only 90 min of non-protective ventilation (*Gonzalez-Lopez et al., 2013*). Heuer et al. reported a non-significant tendency towards more severe hippocampal injury within four hours in pigs with acid-aspiration induced lung damage (*Heuer et al., 2012*). In another porcine two-hit model of acid-aspiration and non-protective ventilation, no signs for hippocampal apoptosis were found (*Bickenbach et al., 2011*). These discrepancies could be explained by the fact, that we counted HE-stained neurons, without differentiation in the morphology of the cells, whereas Heuer et al. evaluated nuclear pyknosis and eosinophilic degeneration of the cytoplasm as markers of neuronal cell damage. Bickenbach et al. used similar criteria, whereas Gonzales-Lopes considered PARP-1 positive neurons as evidence of injury. Hence, we evaluated the proportion of neurons with nuclear pyknosis or hypereosinophilia as markers for acute neuronal damage. We found significantly more eosinophilic and pyknotic neurons in the hippocampus of OAI animals compared to untreated animals whereas the overall number of neurons in the hippocampus remained unchanged.

There are several limitations of our study. Lung injury induced by injection of oleic acid is a well-established model for acute ARDS characterized by a profound change in oxygenation due to microvascular thrombosis, leucocyte infiltration, necrosis and leakage of protein-rich fluid into the airspace with extravascular lung water accumulation (*Goncalves-de Albuquerque et al., 2015*). Our model showed oxygenation impairment and extravascular lung water accumulation, which improved during the course of the study. The wet-to-dry ratio, as a marker for pulmonary edema did not differ between the lung injury group and the control group. This indicates an improvement of the lung injury, possibly due to the lung-protective ventilator setting. Gonzales-Lopez et al. found significant apoptosis in the hippocampus of mice that were non-protectively ventilated, even without ARDS (*Gonzalez-Lopez et al., 2013*). They showed that this hippocampal damage was mediated by the vagal nerve, which is activated by stretch sensors in the lungs (*Gourine et al., 2008*). Our lung-protective ventilation may have prevented vagal nerve mediated hippocampal damage

by reducing distension on the lungs. Furthermore, the observed improvement of lung injury markers could correspond to reduced lung inflammation. This could lead to an attenuated cytokine release in the blood with less pronounced systemic inflammatory response and accordingly reduced cerebral inflammation and apoptosis, compared to septic conditions, which lead to hippocampal damage (*Semmler et al., 2005*) and long-term memory deficits (*Semmler et al., 2007*). Another limitation is the duration of our experiment. Usually patients with ARDS require mechanical ventilation for several days (*Bellani et al., 2016*), whereas our experiment was limited to 18 h. This may be too short to induce a profound cerebral response that is not mediated by the nervous system itself, but by the circulatory system. The systemic immune response usually lacks some days after localized infection. Finally, we only examined mRNA of the cerebral cytokines, not proteins. The increase of mRNA is not necessarily followed by an increase in protein concentration. Accordingly, our results should be interpreted with caution.

## CONCLUSIONS

Hippocampal cytokine mRNA increases within 18 h after induction of acute lung injury without histological evidence of neuronal loss. Furthermore, we found an increase in serum cytokines after induction of lung injury. Although we did not evaluate cognitive function after acute lung injury, prior studies have documented that an increase in hippocampal IL-6 and TNFalpha result in cognitive dysfunction. Therefore, it is plausible that an increase in cytokine mRNA may contribute to cognitive decline in the setting of ARDS, especially if it is followed by an increase in cerebral cytokine synthesis. Further studies should investigate the temporal evolution of cytokine elevation and its influence on cognitive dysfunction.

## ACKNOWLEDGEMENTS

We thank Dagmar Dirvonskis and Dana Pieter for excellent technical support. The authors are grateful to Prof. Elisabeth Jane Rushing (University Hospital Zurich, Switzerland) for critically reading the manuscript. This work contains parts of the medical thesis of KoFo and JS.

### Funding

The authors received no funding for this work.

### Competing Interests

The authors declare there are no competing interests.

### Author Contributions

- Jens Kamuf conceived and designed the experiments, performed the experiments, analyzed the data, prepared figures and/or tables, authored or reviewed drafts of the paper, and approved the final draft.

- Andreas Garcia Bardon, Alexander Ziebart, Konstantin Folkert, Johannes Schwab, Robert Ruemmler, Miriam Renz and Denis Cana performed the experiments, authored or reviewed drafts of the paper, and approved the final draft.
- Katrin Frauenknecht performed the experiments, analyzed the data, authored or reviewed drafts of the paper, and approved the final draft.
- Serge C. Thal conceived and designed the experiments, authored or reviewed drafts of the paper, and approved the final draft.
- Erik K. Hartmann conceived and designed the experiments, analyzed the data, authored or reviewed drafts of the paper, and approved the final draft.

### Animal Ethics

The following information was supplied relating to ethical approvals (i.e., approving body and any reference numbers):

Landesuntersuchungsamt Rheinland-Pfalz, Koblenz, Germany (the institutional and state animal care committee which is responsible for ethical evaluation and surveillance of animal studies) approved this research (G14-1-077).

### Data Availability

The raw data are available in the Supplementary Files.

### Supplemental Information

Supplemental information for this article can be found online at http://dx.doi.org/10.7717/peerj.10471#supplemental-information.

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
