# Peer review of "Experimental lung injury induces cerebral cytokine mRNA production in pigs"

_PeerJ, doi:10.7717/peerj.10471_

## Round 0.1 · original submission · Major Revisions

Please address all comments of reviewers 2 and 3 - they made some very relevant comments regarding data analysis and interpretation.

Reviewer 1 ·

Basic reporting

no comment

Experimental design

no comment

Validity of the findings

no comment

Additional comments

Kamuf et. al. investigate cerebral cytokine production in an experimental lung injury model induced by oleic acid injection in pigs. They found an increase in hippocampal IL-6 mRNA in the OAI group compared to control group without histological evidence of neuronal damage.

- The study is well written with recent literature cited in the text.
- A detailed and critical disscussion is made and limitations of the study are included.

·

Basic reporting

Paper is generally clear throughout.

Please proof read the paper as some minor english corrections need to be made. e.g. Line 102 "were" should be "was"
Line 216: "only a trend towards higher levels of TNFalpha" add "was observed".
Please subscript all of the "2s" in FiO2 throughout the manuscript.

Experimental design

The statistics are appropriate only for the inflammation measurements. Physiological measurements should be examined using a 2-way Repeated Measures ANOVA and a post-hoc test will determine time points that are significant.
Do you mean 10% formaldehyde which is the normal fixation strength (i.e. make 4% paraformaldehyde into 10% formaldehyde solution)?

Validity of the findings

The assessment of microglia conducted was rather simplistic as you simply showed that the number of microglia was not different, which isn't overly surprising, however the state of the microglia, whether it is active or not, is important. If done correctly the histology of the microglia will tell you whether they are ameboid or ramified and that will tell you whether an inflammatory response is taking place within the regions. Further, microglia in inflammatory states often form aggregations and a thorough examination would include the total area of aggregations within the region of interest, % coverage of aggregations, etc. Simply counting will miss these inflammatory signals.
Lines 223-227 - IBA1 stains microglia, whether they are active or not. Did you try to distinguish between them?

Additional comments

Line 261-62: "Even though the pulmonary damage score was significantly
higher in the lung injury group, this indicates an improvement of the lung injury" - this is worded poorly. How can increased injury indicate an improvement of lung injury?
Did you measure any anti-inflammatory cytokines which may explain the reduction in cytokines or injury discussed?

Reviewer 3 ·

Basic reporting

No comment

Experimental design

No comment

Validity of the findings

No comment

Additional comments

Kamuf et al., investigated the mechanisms by which acute lung injury induces cognitive impairment in a pig model of Acute respiratory distress syndrome (ARDS). The authors suggest that acute lung injury induces peripheral inflammation (increasing plasma IL-6 and TNFa) which would mediate an increase of IL-6 and TNFa mRNA synthesis in hippocampus.
The topic of research is interesting and well introduced, the manuscript is clearly written and the used methods are appropriated. However, in my opinion some aspects need to be deepened and better discussed to improve even more the quality of research. These are addressed section by section, below:

UNDER METHODS:
- How long were the brains fixed in 4% formaldehyde solution?
- How were the HE and IBA1 immunohistochemistry quantified? Please, briefly explain it in methods.
- From which body’s part were taken the blood samples?

UNDER RESULTS:
- To study the effects of acute lung injury on neuroinflammation the authors quantified the number of microglial cells in CA2 region of hippocampus but no differences between groups were found. Activation of microglial cells is not necessarily accompanied by proliferation (as the authors also discussed). Thus, in my opinion, to increase even more the quality of research, the authors should quantify eventual morphological changes in microglial cells (perimeter of microglia or their cell body size). Furthermore, considering the small size of CA2 hippocampus region, it would be interesting and more appropriate analysing the number and morphological changes in the whole hippocampus.
- The authors should also perform the morphological/proliferative analysis of microglia in cortex, as they did the content of mRNA cytokines.
- The authors also state: “there were no differences in the number of neurons”. These results are not so unexpected considering the experimental design (animal were sacrificed 18 hours after inducing lung injury). The authors should analyse other parameters to evaluate the neuronal damage such as nuclear pyknosis and eosinophilic degeneration of the cytoplasm as in in Heuer et al., (cited by the authors). The authors should also analyse other hippocampal regions (CA1, CA3, and DG).
- The authors only analysed the content of mRNA cytokines in hippocampus and cortex but not their protein. An increase of mRNA is not necessarily accompanied by an increase of the protein. To increase the quality of their research and better support their conclusions, the authors should also analyse the content of the protein. At least, a note of caution should be added discussing their results.


MINOR CONCERNS
- In my opinion, the main title “Experimental lung injury induces cerebral cytokine production in pigs” is misleading. In fact, the authors only have analysed the content of mRNA but not the protein of cytokines. Please change or support it experimentally by analysing the content of cytokines besides the mRNA.
- Most of graphs are lacking of units of measurement. Please add them where lacking.
- Under figure 1, the authors should also include one representative photo per group of HE stain and IBA1 immunohistochemistry.

---

## Round 0.2 · Minor Revisions

Please provide a response if and how you adressed the minor revision request by Reviewer 3. Based on your response I might make a decision on the manuscript without sending it back to the reviewers.

·

Basic reporting

no comment

Experimental design

no comment

Validity of the findings

no comment

Additional comments

The author has adequate addressed my concerns.

Reviewer 3 ·

Basic reporting

No comment

Experimental design

No comment

Validity of the findings

No comment

Additional comments

The authors have satisfactorily addressed most of the comments but there are still a few aspects of manuscript that in my opinion could be improved to increase even more the quality and the fluidity of the manuscript:

UNDER METHODS:
- The authors should add in the text a sentence about the time of fixation of the brains in 4% formaldehyde solution
- The authors should add a sentence in the text about from which body’s part were taken the blood samples

UNDER RESULTS:
- Why did the authors delete the IBA1 graph from the figure 1? Please, restore it including the new data obtained from the quantification of CA1, CA3, CA4 and DG regions. Please, also add a graph with the quantitative data from the morphological analysis and cite them in the text.

UNDER FIGURES:
- Under Suppl. Figure 1, the authors should also include one representative photo per each analysed group and region (CA1,CA2, CA3, CA4, DG) of HE stain and point some example of pyknotic and eosinophilic degeneration on it.
- Under Suppl. Figure 2, please point some example of activated and resting microglia.

---

## Round 0.3 · accepted · Accept

If possible, please add some legends to your supplemental figures - what do panels A, B, and C show? Please also check for labelling of the scale bar, it is missing at least at Suppl Fig. 2